

# Activity-specific mobility of adults in a rural region of western Kenya

Jessica R. Floyd[1], Joseph Ogola[2], Eric M. Fèvre[2,3], Nicola Wardrop[4], Andrew J. Tatem[1] and Nick W. Ruktanonchai[1]

[1] WorldPop, Department of Geography and Environmental Science, University of Southampton, Southampton, United Kingdom
[2] International Livestock Research Institute, Nairobi, Kenya
[3] Institute of Infection and Global Health, University of Liverpool, Liverpool, United Kingdom
[4] Department for International Development, Glasgow, United Kingdom

## ABSTRACT

Improving rural household access to resources such as markets, schools and healthcare can help alleviate poverty in low-income settings. Current models of geographic accessibility to various resources rarely take individual variation into account due to a lack of appropriate data, yet understanding mobility at an individual level is key to knowing how people access their local resources. Our study used both an activity-specific survey and GPS trackers to evaluate how adults in a rural area of western Kenya accessed local resources. We calculated the travel time and time spent at six different types of resource and compared the GPS and survey data to see how well they matched. We found links between several demographic characteristics and the time spent at different resources, and that the GPS data reflected the survey data well for time spent at some types of resource, but poorly for others. We conclude that demography and activity are important drivers of mobility, and a better understanding of individual variation in mobility could be obtained through the use of GPS trackers on a wider scale.

## INTRODUCTION

Population mobility is a complex process with great importance in many fields across the social and health sciences (*Wesolowski et al., 2012*; *Prothero, 1977*; *Bajardi et al., 2011*). Often, people travel because of specific resource needs or activities such as gathering food and water, livelihood and occupational activities, or accessing healthcare. In low-income settings, travel to these resources can be very time-consuming or expensive meaning that people may forego healthcare, employment, or other resources. As a result, the geographic inaccessibility of vulnerable populations can lead to worse health outcomes, a poorer economic outlook, and can widen spatial inequalities (*Pearce et al., 2008*; *Alegana et al., 2018*; *Macintyre et al., 2019*). In Kenya, resource access is particularly important for poverty reduction in rural populations, as people often have to travel further for resource-related activities than their urban counterparts. It is widely accepted that people in rural areas spend more time accessing resources than people in urban areas, and that this likely contributes

Corresponding author
Jessica R. Floyd, jrf1g15@soton.ac.uk

to poverty in these areas. For example, market access is important for several household activities such as gathering food, selling crop surpluses and buying medicine, and studies have shown that poor market access contributes to poverty in rural areas (*Chamberlin & Jayne, 2013*). Similarly, poor water source access means reduced time for income-generating activities and therefore contributes to household poverty, as well as being linked to poor health (*Whittington, Mu & Roche, 1990*; *Cook, Kimuyu & Whittington, 2016*).

Geographic inaccessibility of healthcare is also a known driver of poor health outcomes, particularly in rural settings (*Noor et al., 2003*; *Ruktanonchai et al., 2016*). Government policies in Kenya have responded accordingly, through measures designed to ensure that everyone lives within 5 km of basic healthcare services. In 2003 it was estimated that 82% of the population live within 5 km of a primary healthcare and referral service (*Noor et al., 2004*). Because geographic accessibility is vitally important for ensuring vulnerable populations can utilise healthcare, a significant body of recent research focuses on modelling accessibility across national scales. Often, these accessibility models assume that people visit their nearest clinic (*Alegana et al., 2012*), using accessibility surfaces or straight-line distances to predict clinics used and associated travel times. In reality, geographic accessibility remains highly heterogenous across the country despite new clinics in resource-poor areas (*Kenya Ministry of Medical Services and Ministry of Public Health & Sanitation, 2013*), and straight-line distance is not the only factor that impacts whether people can access healthcare in a reasonable time without undue expense. Other factors such as poor road quality and a lack of public transport options can severely impact mobility and therefore healthcare access in rural areas where the most vulnerable populations live (*Tanser, Gijsbertsen & Herbst, 2006*; *Airey, 1992*). In Kenya, studies have found that people often visit clinics other than their nearest one, for reasons such as the availability of medicines or the perceived effectiveness of the facility (*Mwabu, 1986*). One study found that only 54–63% of people surveyed visited their nearest facility, with the rest visiting other clinics (*Mwabu, Mwanzia & Liambila, 1995*).

Moreover, resource-related movement is quantitatively different from other types of mobility, so accessibility models that predict general mobility patterns may not accurately reflect resource seeking behaviour. For example, a recent study in Iquitos, Peru found that residents moved significantly further for commercial and familial reasons than for healthcare (*Perkins et al., 2014*). Sociodemographic factors could also influence mobility and resource-specific movement: household income, rural/urban context (*Molyneux et al., 1999*) and gender are all important determinants of mobility, but spatial models of access often lack this demographic information, assuming that all adults access resources such as markets and health facilities identically, regardless of socioeconomic context. More detailed movement datasets could help to improve these models by providing evidence to support or reject these key assumptions and better understand how movements driven by different activities vary.

Additionally, mobility studies typically focus on single types of resource access without comparing against travel of other types (*Schröder et al., 2018*; *Kanuganti et al., 2015*), particularly in a rural context where activity-dependent mobility models could provide a richer picture of how people spend their time, and how resource-specific movement

could exaggerate or mitigate geographic inaccessibility. Traditionally, studies of individual mobility have relied on survey methods, which may be affected by recall bias (*Wesolowski et al., 2015*). In recent years, specialised tools such as personal Global Positioning System (GPS) tracking devices employed at a household level have facilitated the collection of detailed movement information (*Searle et al., 2017*; *Vazquez-Prokopec et al., 2013*; *Parsons et al., 2014*). Personal GPS trackers have been used in urban and rural settings to document the movements of both humans and animals in a variety of contexts, from investigating the diving behaviour of certain species of birds (*Browning et al., 2018*; *Ryan et al., 2004*) to the social structures of cattle in pastoral communities (*Moritz et al., 2012*) to detailed human movements in urban settings (*Vazquez-Prokopec et al., 2013*). With good portability, weight and battery life, this type of tracker has also been used in research into healthcare access (*Siedner et al., 2013*) and vector-borne disease (*Searle et al., 2017*; *Vazquez-Prokopec et al., 2013*). Given the growing populations in rural areas of lower-income countries, detailed knowledge of activity spaces and health facility access in a rural context could improve our understanding of specific types of resource access.

Here, we use surveys of people in a rural area of western Kenya to capture movements to and time spent on different types of activity, and examine links between these and demographic characteristics. We use GPS trackers to explore where the same people spent their time outside of their households. We then compare these two sources of data to see how well they capture proportions of time spent on different types of activity. The results from this study shed light on the activity spaces of people in relation to resource access in a rural setting and provide evidence for how well GPS trackers are able to capture daily activities compared to survey methods. They also help quantify the importance of including sociodemographic and activity-specific movement into geographic accessibility models, particularly for healthcare infrastructure.

## MATERIALS & METHODS

### Study area and population

This study was conducted in the densely populated county of Busia in the Lake Victoria basin region of western Kenya (Fig. 1). Busia county has a population of just under one million people (*Kenya National Bureau of Statistics, 2017*), of whom approximately 80% live in rural areas (as defined by the latest DHS survey) and practice smallholder subsistence farming, mainly operating mixed crop-livestock systems with 60% of households in rural areas owning cattle (*Kenya National Bureau of Statistics, 2015*).

We used a clustered random sampling strategy to select 55 households to visit and survey in Busia county. Of the 181 sublocations in the county, 11 were selected at random and within these we selected five households for participation in the study, by randomly generating coordinates within each sublocation, and choosing the household closest to each of the coordinates, up to a maximum of 200 metres away. A surplus of ten coordinates were generated in each sublocation, so that if a household could not be identified within 200 metres, the next set of coordinates were used. We used QGIS software tools (*QGIS Development Team, 2019*) for the random selection of sublocations and coordinates. Due
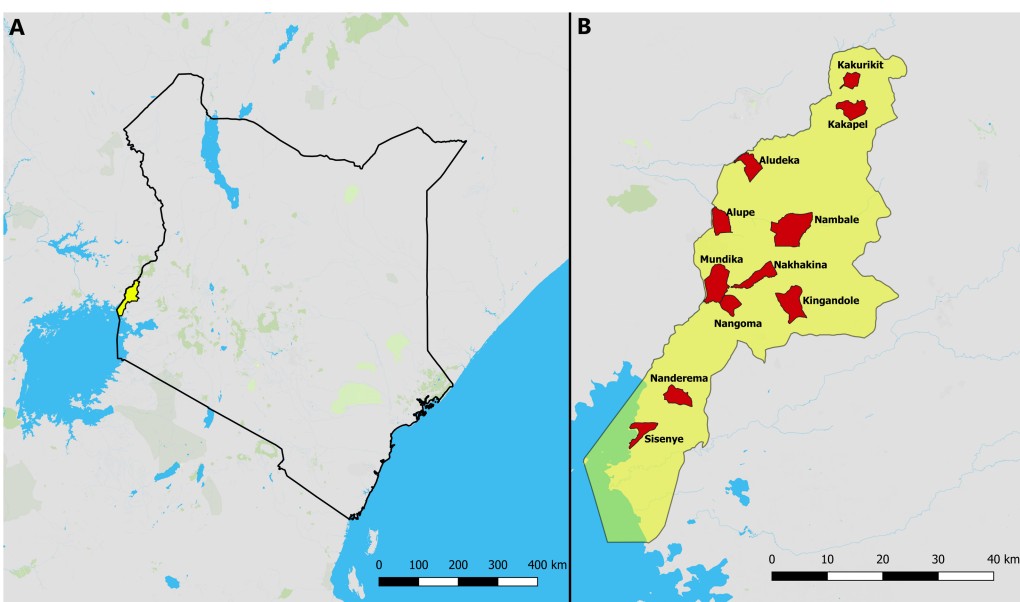

**Figure 1** **Map of the study area.** (A) Kenya with Busia county highlighted in yellow. (B) Busia county with the 11 selected sublocations highlighted in red.

to the lack of appropriate data to base a sample size calculation on, we chose to sample 55 households from a broad geographical range based on the manpower and time available for fieldwork while optimising the use of the limited number of GPS trackers. We selected 30 of the 55 households for GPS tracking because this was the maximum number of people that could be tracked for a full week with the resources available.

Households were selected if they were the main residence of at least one consenting adult present at the time of the visit. Inclusion criteria for participation were consenting adults aged 18 years or over. Some demographic characteristics of the study population are given in Table 1. If the household declined to participate in the study, the next household closest to the coordinates was visited. All adults within a household were selected for participation in the survey, but only the adult who spent most time looking after livestock (determined from survey responses and verbal communication) was selected for participation in the GPS tracking. If the household had no livestock, the head of the household (determined by verbal communication) was selected instead. These choices enabled analysis of the activities of populations that spent time with livestock and therefore may be at higher risk of some zoonotic diseases. This limits the representativeness of the sample population, but we felt this was acceptable given the small sample size.

## Data collection—survey

A structured survey with closed and open questions was administered to all consenting members aged 18 years or over present in the household at the time of visiting (see Supplementary Information). The survey included questions on demographic characteristics, regular movements outside of the household and activities involving livestock. For movements to places outside the household, we asked about the type of

**Table 1  Individual and household characteristics of sample population.**

| Demographic covariates | Number of participants: survey & GPS | Number of participants: GPS only |
|---|---|---|
| **Gender** | | |
| *Male* | 33 (43.4%) | 18 (69.2%) |
| *Female* | 43 (56.6%) | 8 (30.8%) |
| **Age** | | |
| 18–29 | 22 | 7 |
| 30–49 | 26 | 8 |
| 50–69 | 23 | 7 |
| 70+ | 5 | 4 |
| **Main occupation** | | |
| Farming/agriculture | 45 | 18 |
| Hunting | 2 | 2 |
| Trading | 3 | 1 |
| Other | 18 | 4 |
| Unemployed | 8 | 1 |
| **Relative wealth score (PPI Kenya\*) of participant's household** | | |
| Less than 30 | 16 | 9 |
| 30–50 | 24 | 9 |
| 51 or more | 12 | 8 |
| **Ruminant ownership of participant's household** | | |
| No ruminants | 12 | 6 |
| Ruminants | 40 | 20 |

**Notes.**

*PPI Kenya, Poverty Probability Index for Kenya 2011 (*Kenya | PPI [Internet], 2011*).

place visited (e.g., school, water source, place of worship, market etc.), how often the respondents visited, the mode of transport used and time spent travelling, and how long they usually spend there. The types of places were pre-defined based on information from previous studies in this area (*Fèvre et al., 2017*; *Floyd et al., 2019*), and the survey included open-ended questions to identify any other significant types of place. A village elder was present to facilitate introduction to the household and explanation of the study. The survey was written in English and administered through an interpreter. The survey included 10 closed-ended questions from the Poverty Probability Index for Kenya (*Poverty Probability Index, 2011*). The answers to these questions were scored to obtain a basic index of household wealth which was used to compare relative wealth between households. The answers to the survey, including collection of GPS coordinates to determine household location, were collected on a tablet using a custom-designed survey built with OpenDataKit (ODK) (*Brunette et al., 2013*) software and uploaded to a secure server once an internet connection could be established.

## Data collection—GPS data

During the same visit, the consenting participant was given a GPS tracker (i-gotU GT-600 GPS logger; Mobile Action Technology, Inc., Taipei City, Taiwan) to wear for one week, fitted to a lanyard, which could be worn around the neck or carried in a pocket. We chose this length of time based on previous studies (*Bohte & Maat, 2009*; *Stopher, Daigler & Griffith, 2018*) and to maximise the use of our limited number of GPS trackers. The time interval used on the trackers was one minute and the devices were programmed to power off if stationary for two minutes, then turn on again when movement was detected. At the end of the week, the researchers returned to the household to collect the trackers and download the data.

The data collection was conducted in two phases: household survey and GPS tracking in July/August 2016, followed by GPS tracking only in November/December 2016. In phase two, the same households were visited as in phase one and the GPS tracking was repeated with the same participants where possible, in order to capture potential differences in movement patterns during different seasons, henceforth called the short rainy season (July/August 2016) and the dry season (November/December 2016) in accordance with climate classifications for this region of Kenya (*Thuranira-McKeever et al., 2010*). Of the 26 trackers given out in each season, one device was unrecoverable and two suffered battery issues during the week (these weeks were therefore repeated).

## Data analysis

The survey and GPS data were downloaded in .csv format, then cleaned and analysed using R version 3.1.1 software (*R Core Team, 2020*). Erroneous points in the GPS data were identified by their unlikely speeds and deleted using functions from the *trip* (*Sumner, Wotherspoon & Hindell, 2009*) package. A linear interpolation algorithm was then applied to obtain locations at regular intervals. Erroneous points can occur due to changing atmospheric conditions and building obstructions, and accounted for less than 1% of the dataset. We used survey response, GPS points collected in the field and publicly available datasets of health facilities (*Noor et al., 2009*) to group the places where people spent time into six categories: household or residential places;, shops and markets; places of worship; health facilities; places where livestock activities occurred; and places where activities related to water (but not livestock) occurred. A central GPS point was identified for each of the places, and a 25-metre radius around that point was used to determine when that place was visited by a person, defined as 5 min or more spent within that radius. Due to their larger size, a radius of 50 metres was used for market centres.

We conducted univariable analyses using linear mixed models (LMMs) for the three movement measures calculated from survey responses (frequency of visits, time spent travelling and time spent at places). We log transformed these data to ensure an approximately normal distribution and used linear models to examine their relationships with the covariates. Because of the hierarchical nature of the data (individuals nested in clusters, with repeat measurements per individual), we used linear mixed models which had the individual household nested within the sublocation as a random effect, to account for variation between individuals from different households and within different sublocations.
Variables were chosen for analysis in line with those found to be important to movement metrics in previous studies in the area (*Fèvre et al., 2017*; *Floyd et al., 2019*; *Wardrop et al., 2016*).

### Ethics statement

This study was approved by the Institutional Research Ethics Committee and the Institutional Animal Care and Use Committee of the International Livestock Research Institute (IDs: ILRI-IREC2016-11; IACUC-RC2016-14; committees approved by the Kenya National Commission for Science, Technology and Innovation (NACOSTI)) and the Ethics and Research Governance board at the University of Southampton (ID:18984). All participants provided signed, informed consent for their participation in the study and the data were stored securely in accordance with the University of Southampton data storage policy.

## RESULTS

We used the survey data to explore where people spent most of their time, how long they took to get there and how long they stayed when visiting places outside of the household. When examining the places people travelled to regularly, we found substantial variation in the frequency of visits to different types of places compared to healthcare visits (Fig. 2). As expected, participants visited health facilities less often than other types of locations, the modal class being three to six visits a year with very few visiting more often than monthly. Visits to areas where livestock activities occurred (such as grazing pastures) and to water sources were the most polarised out of all types of travel, with most visits occurring daily or not at all.

We asked participants about how long they spent travelling to different places and found that mean one-way travel times to health facilities ranged from four minutes to two hours Participants reported spending the shortest time travelling when tending to livestock (median of nine minutes one way), and the longest times when visiting a market (median of 34 minutes one way). We found the greatest range of travel times for visits to health facilities, markets and places of worship, while visits to livestock areas, water sources and other households tended to have shorter travelling times (Fig. 3).

We also asked about the amount of time people spent at different places. Participants reported spending the longest times at health facilities and places of worship (median of three hours for both), and the shortest times at other households (median of 45 minutes) and water sources (median of nine minutes). Figure 4 shows how the time spent by participants varied by type of place. We used the outcome measures (frequency of visits, time spent travelling and time spent at places) and the demographic characteristics collected in the survey to conduct a univariable analysis to explore associations (Table 2). We found that men reported spending more time at and making more frequent visits to places where livestock activities occurred compared to women ($p = 0.017$ and $p = 0.007$, respectively), while women reported spending more time at and making more frequent visits to health facilities than men did ($p = 0.019$ and $p < 0.001$, respectively). Women also made more frequent visits to water sources than men but did not spend significantly more time there ($p = 0.075$). Older

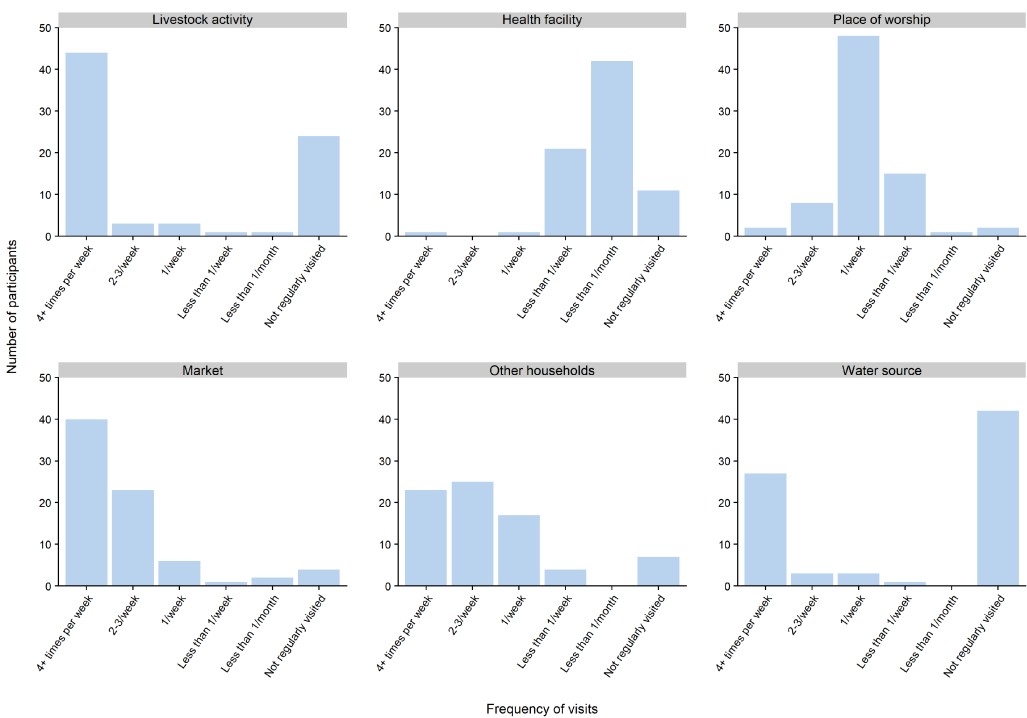

**Figure 2  Frequencies of visits to different types of places.**

people tended to visit health facilities and water sources less often ($p = 0.041$ and $p < 0.001$, respectively) and spend less time at places of worship ($p = 0.003$) than younger people. No significant relationships were observed between time spent travelling to places and any of the demographic characteristics tested; these results are reported in the Supplementary Information.

## GPS validation

Lastly, we used the GPS data to explore the time participants spent outside of their households, and measure how variable overall movements were in our study population. We found that people spent between 5% and 52% of their time outside of their households over the week that they were tracked. For each GPS dataset, we were able to identify the type of place visited for between 2% and 97% of the time recorded outside of the household; some movements could not be characterised because not all locations in the local area were identifiable through the survey or mapping techniques. We selected GPS datasets where the season tracked was the same as when the survey was conducted and where over a third of minutes were identified, resulting in 27 GPS datasets with matching survey data. We compared these to explore how well the GPS data were able to mirror how people spent their time at different places. For this comparison, we only selected types of places that at least 50% of people with GPS data visited at least once a week, apart from the health facility locations, which we included for their public health importance. Unsurprisingly since not all GPS points could be identified, participants reported spending more time at all place
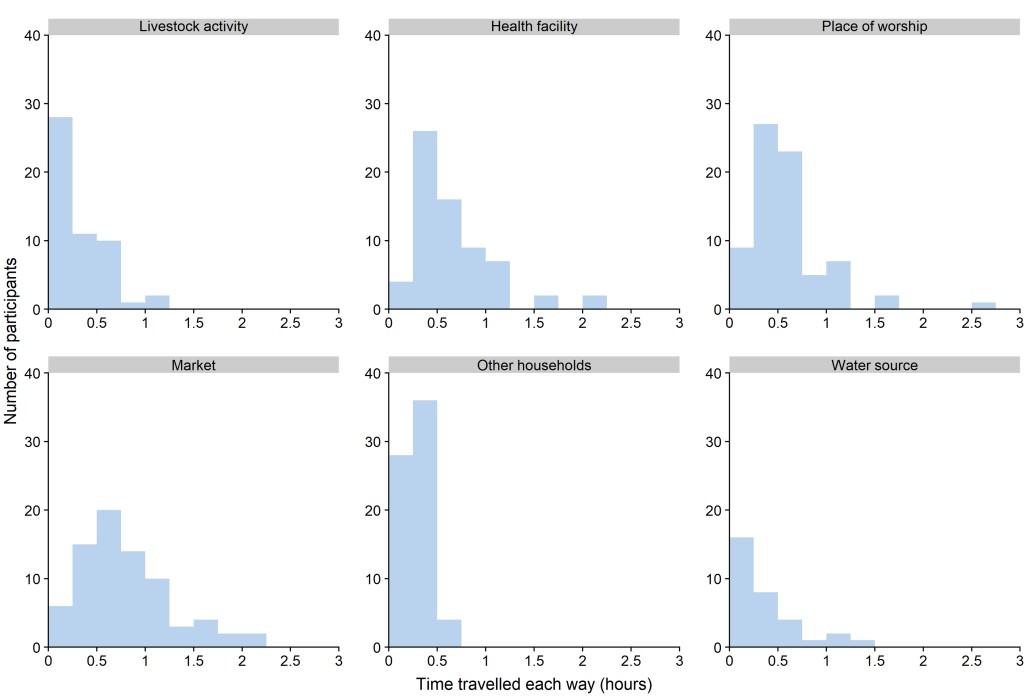

**Figure 3** Time spent travelling one-way to different types of places.

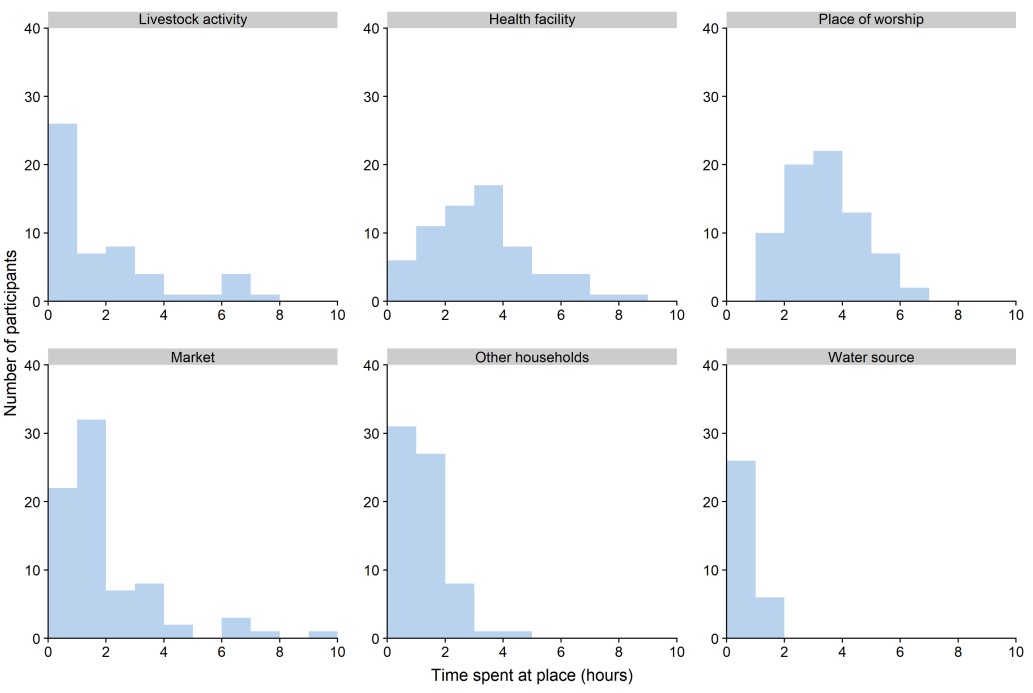

**Figure 4** Time spent by participants at different types of places.

**Table 2** **Relationships between movement metrics and demographic characteristics of the surveyed population for different types of activity.** Univariable analysis of movement metrics using beta regression in a linear mixed model. Estimates are given with 95% confidence intervals in square brackets.

| Activity type | | Explanatory variable | Estimate | *p*-value |
|---|---|---|---|---|
| Livestock activity | Visits per week | Gender: male [Ref = female] | 15.78 [2.10, 109.68] | 0.007** |
| | | Occupation: non-farmer [Ref = farmer] | 0.08 [0.01, 0.65] | 0.022* |
| | | Household wealth | 0.17 [0.01, 3.69] | 0.263 |
| | | Age | 1.04 [0.97, 1.11] | 0.245 |
| | Time spent | Gender: male [Ref = female] | 5.33 [1.41, 20.22] | 0.017* |
| | | Occupation: non-farmer [Ref = farmer] | 0.68 [0.17, 2.79] | 0.591 |
| | | Household wealth | 5.46 [0.95, 31.05] | 0.063 |
| | | Age | 1.03 [0.99, 1.07] | 0.206 |
| Health facility | Visits per week | Gender: male [Ref = female] | 0.12 [0.04, 0.36] | <0.001*** |
| | | Occupation: non-farmer [Ref = farmer] | 2.96 [0.93, 9.44] | 0.069 |
| | | Household wealth | 0.25 [0.06, 1.10] | 0.071 |
| | | Age | 0.97 [0.93, 1.00] | 0.041* |
| | Time spent | Gender: male [Ref = female] | 0.65 [0.46, 0.92] | 0.019* |
| | | Occupation: non-farmer [Ref = farmer] | 0.88 [0.60, 1.28] | 0.504 |
| | | Household wealth | 0.70 [0.42, 1.18] | 0.193 |
| | | Age | 0.99 [0.98, 1.01] | 0.338 |
| Place of worship | Visits per week | Gender: male [Ref = female] | 0.50 [0.24, 1.00] | 0.054 |
| | | Occupation: non-farmer [Ref = farmer] | 1.01 [0.48, 2.14] | 0.984 |
| | | Household wealth | 0.65 [0.23, 1.84] | 0.425 |
| | | Age | 1.01 [0.98, 1.03] | 0.617 |
| | Time spent | Gender: male [Ref = female] | 0.81 [0.65, 1.00] | 0.051 |
| | | Occupation: non-farmer [Ref = farmer] | 1.03 [0.82, 1.28] | 0.812 |
| | | Household wealth | 0.72 [0.54, 0.96] | 0.032* |
| | | Age | 0.99 [0.98, 1.00] | 0.003** |
| Market | Visits per week | Gender: male [Ref = female] | 1.09 [0.37, 3.09] | 0.875 |
| | | Occupation: non-farmer [Ref = farmer] | 0.43 [0.14, 1.36] | 0.135 |
| | | Household wealth | 1.29 [0.27, 6.13] | 0.753 |
| | | Age | 0.97 [0.95, 1.00] | 0.121 |
| | Time spent | Gender: male [Ref = female] | 1.24 [0.54, 2.87] | 0.611 |
| | | Occupation: non-farmer [Ref = farmer] | 0.64 [0.28, 1.47] | 0.299 |
| | | Household wealth | 0.98 [0.34, 2.84] | 0.970 |
| | | Age | 0.99 [0.96, 1.01] | 0.316 |
| Household visits | Visits per week | Gender: male [Ref = female] | 1.39 [0.38, 4.85] | 0.612 |
| | | Occupation: non-farmer [Ref = farmer] | 0.92 [0.24, 3.55] | 0.909 |
| | | Household wealth | 1.12 [0.16, 7.49] | 0.908 |
| | | Age | 0.97 [0.93, 1.01] | 0.122 |
| | Time spent | Gender: male [Ref = female] | 1.22 [0.78, 1.93] | 0.387 |
| | | Occupation: non-farmer [Ref = farmer] | 0.83 [0.51, 1.33] | 0.438 |
| | | Household wealth | 0.87 [0.46, 1.62] | 0.655 |
| | | Age | 1.00 [0.98, 1.01] | 0.663 |

**Table 2 (*continued*)**

| Activity type | | Explanatory variable | Estimate | *p*-value |
|---|---|---|---|---|
| Water activity | Visits per week | Gender: male [Ref = female] | 0.02 [0.00, 0.12] | <0.001 *** |
| | | Occupation: non-farmer [Ref = farmer] | 4.80 [0.47, 49.31] | 0.191 |
| | | Household wealth | 0.10 [0.00, 2.66] | 0.176 |
| | | Age | 0.87 [0.81, 0.92] | <0.001 *** |
| | Time spent | Gender: male [Ref = female] | 2.52 [0.92, 6.65] | 0.075 |
| | | Occupation: non-farmer [Ref = farmer] | 0.86 [0.34, 2.13] | 0.755 |
| | | Household wealth | 0.69 [0.22, 2.17] | 0.535 |
| | | Age | 1.00 [0.97, 1.03] | 0.976 |

**Notes.**
*$p < 0.05$.
**$p < 0.01$.
***$p < 0.001$.

types than captured by the GPS (hours reported in survey minus hours recorded by GPS was positive). In most cases, this difference was less than one hour of difference per day for each type of place.

Because the GPS data consistently underreported the absolute amount of time spent at different places compared to the survey data, we also compared how well the survey and GPS data were able to capture the proportions of time spent at the different place types. Figure 5 shows how GPS data found very similar proportions of time spent compared to those reported in the survey for some types of places, such as shops and places where livestock-related activities occurred. Other types of places had large differences in proportion of times spent between the two datasets—specifically, time spent at places of worship and visits to other households and health facilities. Using a paired $t$-test we found significant differences in the mean proportions of time spent at other households and places of worship ($p = 0.002$ and $p = 0.041$, respectively) between the survey and GPS datasets. The GPS datasets recorded more trips to other households than recounted in the survey, while the survey had more trips to places of worship than recorded in the GPS, potentially due to a recall bias effect: people may be more likely to recall movements that they place higher intrinsic value on. There is also the possibility of a 'social desirability' bias: participants may be tempted to overstate the amount of time spent at more desirable locations such as places of worship, particularly as the survey was conducted in the presence of village elders.

## DISCUSSION

Geographic inaccessibility is a primary factor for many poor communities that are unable to easily use important infrastructure and natural resources, particularly in rural areas (*Pearce et al., 2008*; *Chamberlin & Jayne, 2013*; *Clark, Gertler & Feldman, 2003*). Even within these communities, accessibility can vary between individuals and can manifest differently when people are traveling for various types of resources (*Perkins et al., 2014*; *Schröder et al., 2018*). Better understanding of how accessibility varies with type of activity and with an individual's socioeconomic context could help inform future analyses on geographic accessibility. This socioeconomic and activity-based understanding of mobility can also

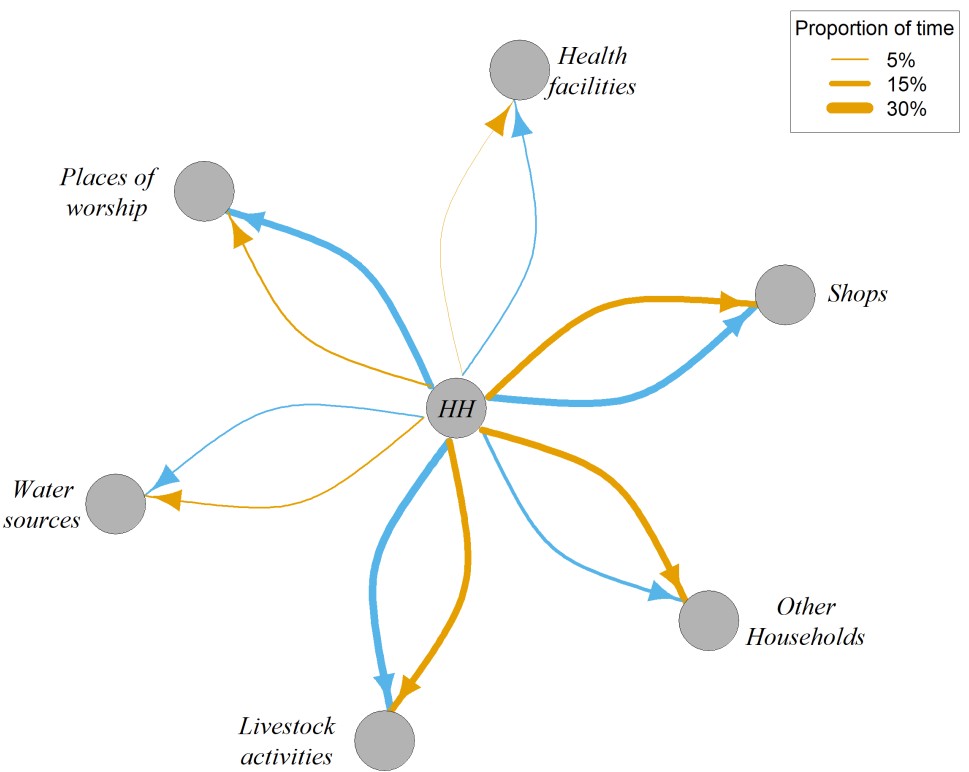

**Figure 5   Network representation of mean proportions of time spent in different types of places.** Mean proportions of time spent from GPS (orange) and survey (blue) data. HH = Household.

help with identifying populations within communities at especially high risk of being unable to access healthcare and other essential services.

When comparing accessibility to a variety of resources we found that the time people spent accessing different resources was not homogenous and had some links to demographic characteristics. Specifically, we observed that mobility related to livestock-related activities and health facilities were correlated with factors such as occupation, age and gender, while other types of activities like household visits and market visits were not. The small sample size and bias towards rural households mean that the representativeness of these results is limited, but nevertheless, combined with evidence of social differences in resource access from previous studies in this area (*Okwi et al., 2007*), they suggest further research into inequalities in resource access could be beneficial for improving individual and thus population welfare.

Broadly, we also found that people access different resources with variable frequencies and spent varying amounts of time there. For example, people visited water sources and places where livestock-related activities occured either frequently or not at all, while other types of places like health facilities and places of worship were visited by most participants on a regular basis. The time spent at each type of place also showed large variation, with people spending the longest times at health facilities and places of worship, and the shortest times at other households and water sources. These findings, while unsurprising, demonstrate

a heterogeneity in resource access between individuals. Travel times to health facilities in our study were longer than to other types of places—a finding which differs from results in Iquitos, Peru, where people travelled less far for health reasons than for commercial and household reasons (*Perkins et al., 2014*). This highlights how health facilities in rural communities such as our study area are likely more inaccessible than other resources when compared to access in an urban area such as Iquitos.

Notably, when comparing health facility visits and water source visits (two resources widely regarded as fundamental human rights), we observed that people spent longer both travelling to and at health facilities compared to water sources. However, travelling times to health facilities suggest that most people live within the government target of 5km from a health facility, and previous evidence from rural Kenya found that travel times to health facilities did not affect child mortality in areas with many facilities (*Moïsi et al., 2010*), leading to calls for more focus on social determinants and quality of care and other factors that influence healthcare access and use. Our findings underscore this, as people reported having to spend a mean of three hours waiting at health facilities, compared to less than 10 minutes to access water.

We also found some significant gender disparities in the amount of time people spent at different places, which could influence healthcare utilisation patterns. Men reported spending longer than women on livestock-related activities, while women spent longer than men at health facilities and water sources. Women reported visiting health facilities more frequently than men, possibly because they are usually responsible for their children's healthcare as well as their own (*Paolisso, Baksh & Thomas, 1989*). The differences in time spent by gender suggests that women are having to wait longer to access healthcare than their male counterparts. Previous studies in rural communities in Kenya and neighbouring Tanzania have found that waiting times, like time spent travelling to the facility, are a major barrier to accessing care and can result in delayed care-seeking behaviour, particularly for women due to the opportunity cost of accessing care over their domestic responsibilities (*Thaddeus' & Maine, 1994*; *Mason et al., 2015*; *Mubyazi et al., 2010*). Since women visit health facilities more often on behalf of their families and spend more time there, measures to reduce waiting times could have a direct benefit to both their and their families' health.

Finally, we compared the travel times and frequencies reported in GPS and survey data, to quantify potential biases in each and address the value of both datasets. The survey results are likely affected by recall bias and a social desirability bias. Notably, the GPS data were limited by the short data collection period and the small sample size, failing to capture visits to health facilities in the time available. Nevertheless, for places frequently visited during the data collection period, they more objectively reflected proportions of time spent there. Compared to surveys, GPS tracking technology could give a more complete picture of activity spaces in rural contexts, but requires long data collection periods and a thorough knowledge of the local area. This knowledge could be obtained through a variety of methods, such as a participatory mapping approach, and could help to reduce recall bias effects. Future studies may be able to overcome the time and cost issues of increasing the data collection period by utilising smartphone technology, which already captures the GPS locations of individuals under certain conditions. The collection of smartphone data

in combination with GPS trackers has recently been piloted in an urban area of the UK and demonstrated that location histories from smartphones can be valid datasets for exploring individual movements (*Ruktanonchai et al., 2018*). In low-income areas of sub-Saharan Africa, this method would have a substantial bias towards wealthy people and is dependent on reliable cell phone network coverage, but smartphone ownership and demand for data plans have both been increasing in recent years (*World Bank, 2012*), making this a viable option for similar studies in the future.

## CONCLUSIONS

These results suggest that demography and activity are important drivers of mobility, influencing how scientists should quantify geographic accessibility to resources such as health facilities. Because different types of mobility manifest in different ways and occur with various frequencies, data such as GPS and survey data could be used to quantify mobility for specific types of activities, taking into account the advantages of each. Since healthcare-related mobility was particularly time-consuming and appears to be linked to several demographic characteristics in our small study, further research in this area could shed light on how people in different demographic and socioeconomic contexts access healthcare and therefore help to improve access in low-income settings.

## ACKNOWLEDGEMENTS

The authors are grateful to the county commissioner, the chiefs, assistant chiefs and village elders of Busia County, Kenya for facilitating the fieldwork. We would also like to thank the people of Busia County, Kenya for participating in the study.

### Funding

This project was supported by the Department of Geography and Environmental Sciences, University of Southampton, and by the Biotechnology and Biological Sciences Research Council, the Department for International Development, the Economic & Social Research Council, the Medical Research Council, the Natural Environment Research Council and the Defence Science & Technology Laboratory, under the Zoonoses and Emerging Livestock Systems (ZELS) programme, grant reference BB/L019019/1. It also received support from the CGIAR Research Program on Agriculture for Nutrition and Health (A4NH), led by the International Food Policy Research Institute (IFPRI). The funders had no role in study design, data collection and analysis, decision to publish, or preparation of the manuscript.

### Grant Disclosures

The following grant information was disclosed by the authors:
Department of Geography and Environmental Sciences, University of Southampton.
Biotechnology and Biological Sciences Research Council.
Department for International Development.

Economic & Social Research Council.
Medical Research Council.
Natural Environment Research Council.
Defence Science & Technology Laboratory, under the Zoonoses and Emerging Livestock Systems (ZELS) programme: BB/L019019/1.
CGIAR Research Program on Agriculture for Nutrition and Health (A4NH), led by the International Food Policy Research Institute (IFPRI).

## Competing Interests

The authors declare there are no competing interests.

## Author Contributions

- Jessica R. Floyd conceived and designed the experiments, performed the experiments, analyzed the data, prepared figures and/or tables, authored or reviewed drafts of the paper, and approved the final draft.
- Joseph Ogola conceived and designed the experiments, performed the experiments, analyzed the data, authored or reviewed drafts of the paper, and approved the final draft.
- Eric M. Fèvre conceived and designed the experiments, performed the experiments, authored or reviewed drafts of the paper, and approved the final draft.
- Nicola Wardrop and Andrew J. Tatem conceived and designed the experiments, authored or reviewed drafts of the paper, and approved the final draft.
- Nick W. Ruktanonchai conceived and designed the experiments, analyzed the data, prepared figures and/or tables, authored or reviewed drafts of the paper, and approved the final draft.

## Human Ethics

The following information was supplied relating to ethical approvals (i.e., approving body and any reference numbers):

This study was approved by the Institutional Research Ethics Committee and the Institutional Animal Care and Use Committee of the International Livestock Research Institute (IDs: ILRI-IREC2016-11; IACUC-RC2016-14; committees approved by the Kenya National Commission for Science, Technology and Innovation (NACOSTI)) and the Ethics and Research Governance board at the University of Southampton (ID:18984).

## Data Availability

Due to the detailed nature of the survey and the small sample size, these data are not published with the article. We have not provided raw GPS data because it is not possible to remove the personally identifying information in these data. The owner of the data can be contacted at jrf1g15@soton.ac.uk.

## Supplemental Information

Supplemental information for this article can be found online at http://dx.doi.org/10.7717/peerj.8798#supplemental-information.

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
