# Peer review of "Activity-specific mobility of adults in a rural region of western Kenya"

_PeerJ, doi:10.7717/peerj.8798_

## Round 0.1 · original submission · Major Revisions

Thanks for this interesting paper. While the reviewers have found the analysis to be technically sound, they have raised numerous queries and made many helpful suggestions for improvement of the manuscript - I consider that these changes will strengthen the paper greatly, particularly since many will help clarify your research.

Reviewer 1 ·

Basic reporting

I found the paper is well organized, easy to understand. Studying mobility in relation to accessibility and quality of life is an important topic and fit with the journal scope.

Experimental design

The author experimental design is acceptable: using survey and GPS tracking devices for two one-week periods.

Validity of the findings

Data analysis, however, remain limited and could be further improved with more visualizations. The key maps that readers would like to see are of all related facilities (i.e. farms, health care offices, water sources, markets, neighborhoods/villages, places of worship, etc.); of participant’s homes; of participant’s GPS tracking positions. This help to bring the analysis to a much finer scale. Visualization could be for all participants or groups of specific characteristics.

Indeed, arguments for accessibility is weak without knowing where all the facilities are in relation to the survey participants’ houses. Maps help to bring in the spatial perspectives in term of how far these are away from houses, what routes were taken, and if one would choose a facility because of preference over the shortest distance.

Reviewer 2 ·

Basic reporting

No comment, everything looked great!

Experimental design

The experimental design and layout was sound.

Validity of the findings

The validity of the findings were sound and the results matched the study design.

Additional comments

PeerJ Manuscript #38955

Activity-specific mobility of adults in a rural region of western Kenya

Thank you for giving me the opportunity to review this very interesting manuscript. The methodology is appropriate and the findings are strong. The authors found that there were demographic differences in activity and mobility within communities that could be further explored specifically in terms of access to healthcare. They used survey collection methods and GPS data loggers to measure activity space and compared these two methods, which was quite interesting. I manuscript as written is very stong and has public health impacts especially in the areas of healthcare and resource accessibility in rural sub-Saharan African communities. I have no large critiques of the manuscript but several smaller comments to be addressed.

Comments

Methods:

Is a single week representative of regular movement patterns? Why was this time unit chosen?

What is the reasoning behind selecting the household member who spent most time looking after livestock selected? Could this have biased the responses on the survey and generalizability of the GPS data?

What was the time interval used for GPS location collection? Were the devices pre-programmed to power on and off at specific time intervals or collecting constantly?

Were any of the GPS units unable to be recovered from participants or any evidence that participants didn’t wear/forgot to wear the GPS unit during data collection?

For the survey data, were the locations pre-defined and generic or were they based on pilot data and maps of the study area?

Results:
How far back were visits assessed in the survey? It looks like visits per month or per year but the GPS data was for 1 week. How were these differences in time period reconciled?

A range of 2-97% of the time with a known location visited outside the home with the GPS is a wide range. Why was some of the movement not able to be characterized? How were these locations obtained and categorized?

27 of how many GPS datasets were there with matching survey data?

It’s not clear why all points captured by the GPS could not be assigned a location that corresponded with the survey data. This is why it would be helpful to know the capture interval that was used to calculate time spent in locations.

It is difficult to assess accuracy comparing the survey and GPS as neither would be considered the gold standard for movement and activity space data. However, one is an objective measure and one is subjective. In the manuscript the authors are using the survey as a the gold standard measure and comparing the accuracy of the GPS data to that without explaining the data collection intervals and potential errors in the collection of data in known locations. For example, time spent in places of worship and frequency of visits to places of worship were greater than the GPS collected, but the authors not that recall bias of participants more likely to recall these if they have more meaning, but neglect the potential for social desirability bias. As the GPS is objectively collecting these data they may be more accurate than reported as participants may report these movements more frequently as they may be seen more positively by others compared to more frequent visits to other households for socializing which they may under report by comparison. 

Discussion

In the discussion the authors represent the GPS data as the more accurate compared to the survey data, which is stated differently in their presentation of the results. This discrepancy should be corrected. The recall bias and social desirability bias lay within the survey data and not the GPS data which is the more objective measure. 

Smartphone data would be biased towards wealthy people in this area which is correct, though smartphones are becoming more numerous in many SSA settings. The larger problem with smartphone GPS collection is that many of them rely on the cell towers and cell reception rather than GPS satellite technology so they also require data plans and a reliable cell phone network. 

Conclusion

The demographic differences in healthcare and water access are really interesting. I think one of the main conclusions of this study that may not be highlighted in the conclusions is the amount of time spent at healthcare facilities found in the surveys. The accessibility not only involves the distance and time traveled, but the wait time and time spent there. This is another angle that could use more research and can inform policy.

Reviewer 3 ·

Basic reporting

The article uses clear and professional English that is technically correct. The introduction includes relevant prior literature and an appropriate level of background information to couch the analysis in the broader field of knowledge on the topic of geographic access to resources, and questions of rural mobility. The structure of the article sections is standard. The research herein is self-contained and an appropriate unit of publication with reporting that does not extend beyond the reach of the hypothesis.

I have one small note for the introductory material - I suggest changing the instances of the word “affect” to a term that doesn’t so definitively imply causal relationships , for example “influence”, “shape”, “guide”, “impact” etc.

While the figures are relevant to the content of the article, I have a number of suggestions to improve them in consideration of clarity for the reader:

Figure 2: while the frequency charts give an overview of the visit frequencies for different activities, labeling the charts grey bar that matches the rest of the bars in color and orientation is visually confusing for the reader, especially for the top two rows of sub-charts where it’s not immediately obvious that the labels extend beyond the x-axis labels. I suggest labeling each sub-chart along the top. If the authors agree with this change, I further recommend changing subsequent figures to match this labeling style to maintain consistency (as is the current state with the vertical right hand side labels).

Figure 3: I suggest reducing the size x-axis labels so that it a) doesn’t overwhelm the visual content of the chart and for fig. 3 in particular b) so that it is more readable (you could also use the 45 degree labeling style for the duration labels on fig. 3). There is also something confusing about the binning with ½ hours as reference labels – one suggestion is to clearly explain the binning (e.g., 5 minute intervals or whatever it is) in the article text where the figure is discussed, and to consider larger bins for better interpretability (even though the tradeoff is obviously coarser time representation).

Experimental design

Line 202 in the Results section: “We used the survey data to explore where people spent most of their time, how long they took to get there and how long they stayed when visiting places outside of the household.”, “we used the GPS data to explore the time participants spent outside of their households, and measure how variable overall movements were in our study population” (238), “ and “we also compared how well the survey and GPS data were able to capture the proportions of time spent at the different place types.” (252). These are all relevant and meaningful questions that can be answered with the data that were collected, however in my reading, these statements reflect aims that were not clearly stated in the introduction which reads “we use surveys and GPS trackers to compare movements to different types of activity for people in a rural area of western Kenya, including examining the links between resource access and demographic characteristic.” (line 98) I suggest reworking the final paragraph of the introduction to better define the research questions and aims of the research and very clearly state for the reader what they can expect you to report.
The descriptions of the data collection approaches were reasonably thorough and understandable, though I have a number of comments and suggestions relevant to the methods description and detail in the “Data analysis” section and beyond:
Line 174 “publicly available datasets” what datasets spsecifically? This should be included in data section before analysis.
Line 177 “A central GPS point was identified for each of the places, and a 25-metre radius around that point was used to determine when that place was visited by a person, defined as 5 minutes or more spent within that radius. Due to their larger size, a radius of 50 metres was used for town centres.” First, I am confused why town centres are mentioned as a grouping category (line 175) but then are never mentioned again in the text/results. Further, this could absolutely be a case of me misunderstanding the approach here and/or the rural context, but to me it doesn’t make sense to have a larger radius in town centres if you are using this buffer region to determine places of interest (POIs) that people visited – namely because I am picturing a “town centre” as an area that would have more potential POIs clustered more closely together. Is this not how the area is structured? If there are more POIs of different types (e.g., a health center and a place of worship in close proximity) then it would make sense to structure buffer radii such that they are sufficiently separate within the town centre.

At Line 182 you state, “We conducted univariable analyses using linear mixed models” and then in the title for Table 1 the description states “Univariable analysis of movement metrics using beta regression in a generalized linear model.” LMM sounds like an appropriate approach given the nested nature of the data (as explained), but beta regression in GLM is totally different – so which is it? Further, beta regression (in my understanding) uses proportional data as the outcome variable so what exactly would the proportion(s) tested be here if it was used (the results in Table 1 look like LMM to me)?

I understand the univariable approach singling out each location, but did you perform any analysis that included type of location as an explanatory control? It seems to me you could run one model on each outcome variable with all of your variables, including location type, as controls. If you didn’t try that, it’s worthwhile to explain in the methods section why you analysed each location type in isolation.

In terms of explanatory variables, many fewer variables are used in the models than what you gathered – how did you choose what to use and ignore? There should be some brief discussion that justifies your choices.

Overall, I suggest greater detail and clarity on the specific methods employed and justification for the approach.

Validity of the findings

This research is a worthwhile undertaking, and much of the reported results are relevant and appropriate to the (hopefully restated) aims of the paper. I have some suggestions to improve the rigor of the reporting:
A table of summary statistics for the demographics breakdown of respondents/participants is necessary to get an overview of your sample population. Also I would like to know how many participants were the livestock carers vs. the head of household w/ no livestock – surely this would come into play for the frequency charts at the very least when Livestock activity and water source might be influenced by those roles?
Line 224/table 1 – freq of visits, time spent traveling, time spent at places…if time spent traveling wasn’t significant when tested against the demographic characteristics, the non-significant results should still be reported. There is also the issue of the missing town centre place category.

Regarding GPS validation:
251 “Because the GPS data consistently underreported the absolute amount of time spent…” If you state in the background that surveys are subject to recall bias, and then state that an aim of this research is to compare surveys to GPS, this phrasing is problematic because you can’t say definitively where the discrepancy in time reporting occurs between survey or GPS. This fix could be something as simple as “…the GPS data consistently underreported the absolute amount of time spent compared to the survey reporting..” though take care to not overstate the quality and content of either survey or GPS because both are problematic in their own ways.

Fig. 5 and related discussion– line 256 “Other types of places were less accurately captured by the GPS data – specifically, time spent at places of worship and visits to other households and health facilities. Using a paired t-test we found significant differences in the mean proportions of time spent at other households and places of worship (p = 0.002 and p = 0.041, respectively) between the survey and GPS datasets.” It’s suspicious that places of worship, households and health facilities have poor correspondence between survey and GPS results as these also appear to be the only indoor locations that were studied (I’m assuming shops/markets are primarily time spent outdoors or around structures that wouldn’t completely interfere with GPS reception). Locations where you spend time indoors are likely to be subject to the most GPS outage/interference – how can you be sure that the result you found isn’t just a discovery of where GPS works (outside) and doesn’t (indoors)? I would suggest first, explaining in more detail how you handled GPS outages if they were present in the data. I see the linear interpolation for erroneous points that were deleted, but were gaps also filled this way? Perhaps gaps still accurately reflect the time spent in a location even if there isn’t a point at the expected resolution within the gap – either way, it needs to be explained.

Conclusions in the discussion are well-stated for the given results.

---

## Round 0.2 · accepted · Accept

Thank you for addressing the issues raised by the reviewers. I have read through the manuscript again and it is much improved by these points of clarification. Congratulations on an excellent study and paper.

Reviewer 1 ·

Basic reporting

I have no issue with how the paper is written or formatted.

Experimental design

The experimental design is acceptable.

Validity of the findings

Findings are valid based on the analysis methods presented.

Additional comments

I recommended accepting to publish the paper as is. I understand the limitation in providing maps as the author discussed due to privacy concerns for participants. Also, the authors responded well to the comments of other reviewers and improved as the original version.

Reviewer 3 ·

Basic reporting

No comment

Experimental design

No comment

Validity of the findings

No comment

Additional comments

The authors have satisfactorily addressed my concerns with their revisions.